# OVERPARAMETERIZED NEURAL NETWORKS CAN IMPLEMENT ASSOCIATIVE MEMORY

## ABSTRACT

Identifying computational mechanisms for memorization and retrieval is a long-standing problem at the intersection of machine learning and neuroscience. In this work, we demonstrate empirically that overparameterized deep neural networks trained using standard optimization methods provide a mechanism for memorization and retrieval of real-valued data. In particular, we show that overparameterized autoencoders store training examples as attractors, and thus, can be viewed as implementations of associative memory with the retrieval mechanism given by iterating the map. We study this phenomenon under a variety of common architectures and optimization methods and construct a network that can recall 500 real-valued images without any apparent spurious attractor states. Lastly, we demonstrate how the same mechanism allows encoding sequences, including movies and audio, instead of individual examples. Interestingly, this appears to provide an even more efficient mechanism for storage and retrieval than autoencoding single instances.

## 1 INTRODUCTION

Training neural networks to act as a model of associative memory is a problem dating back to the introduction of Hopfield networks (Little, 1974; Hopfield, 1982). Hopfield networks are able to store binary training patterns as attractive fixed points; thus, the network allows for recall of the training patterns given corrupted inputs. In this work, we demonstrate that modern overparameterized deep neural networks trained using standard optimization methods also provide a method for memorization and retrieval by storing training instances as attractors. Moreover, we show that through autoencoding and sequence encoding, the training examples can be recovered from random inputs to the trained network. We are not aware of any observation of this phenomenon in the literature, with the exception of Zhang et al. (2019), which studies the case of memorizing a single image.

**Autoencoding.** We begin with the example of an autoencoder neural network implementing a family of continuous functions in the space $\mathcal{F} = \{f : \mathbb{R}^d \to \mathbb{R}^d\}$. Given training examples $\{x^{(i)}\}_{i=1}^n$, gradient descent is used to minimize the following autoencoder objective:

$$\arg \min_{f \in \mathcal{F}} \sum_{i=1}^n \|f(x^{(i)}) - x^{(i)}\|_2^2$$

After training in the overparameterized regime, we obtain a function $f$ satisfying $f(x^{(i)}) \approx x^{(i)}$ for all $i$, i.e., the function *interpolates* the training images. In Figure 1a, we analyze the function that is learned after training a fully connected autoencoder (architecture detailed in Appendix B) on 10 examples from the CIFAR10 dataset (Krizhevsky, 2009). Note that inputting random images causes the autoencoder to produce outputs that are visually similar to training examples.

While the ability to recall training examples from random inputs can be inspected visually as in Figure 1a, we can also quantify the number of training examples that can be recalled by viewing autoencoders from a dynamical systems perspective. Namely, since an autoencoder is a map from the feature space to itself, we can iteratively apply the autoencoder to the output starting from any input and investigate whether the sequence of iterates converges to a training example. A training example is called an *attractor*, when there exists a set of instances with non-zero measure for which iterating the trained autoencoder map converges to the training example. This happens if the largest

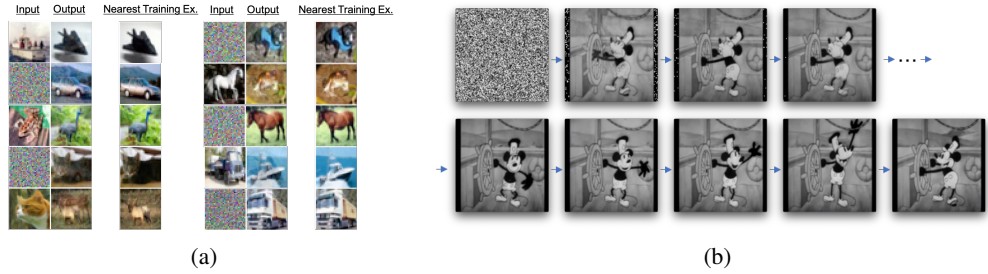

(a)                                        (b)

Figure 1: (a) Fully connected autoencoder trained on 10 images from CIFAR10. When fed Gaussian noise or test images from CIFAR10 the model outputs an image that is visually indistinguishable from a training image. (b) A fully connected network trained to encode a sequence of 389 frames of size $128 \times 128$ from the Disney film "Steamboat Willie." Inputting Gaussian noise and iterating the network yields the entire video.

eigenvalue (in absolute value) of the Jacobian of the autoencoding map at the training example is less than 1 (see Section 2). While the network trained in Figure 1a already outputs an image that is visually indistinguishable from a training example in one iteration, further iterations of the autoencoder result in increasingly more accurate reconstructions of the training examples (see Figure 2a).

As we demonstrate throughout this paper, overparameterized autoencoders store training examples as attractors, and so training examples can be recovered through iterating a trained autoencoder on random input. We note that for a fixed point to be an attractor, all eigenvalues of the Jacobian matrix do not exceed one in absolute value. This is a highly restrictive condition as the number of these eigenvalues is equal to the dimension of the space. Thus, we do not expect a fixed point of an arbitrary high-dimensional map to be an attractor. In that regard, the fact that deep neural networks produce attractors so readily is rather surprising. Moreover, even if an autoencoder stores training examples as attractors, there could be additional spurious attractors outside the training set. Interestingly, in this paper, we present several examples of autoencoders for which we could not discover any attractors outside the training set through iteration from random instances. In particular, we now provide an example of a fully connected autoencoder that stores 500 real-valued training examples as attractors with no other observable attractors.

**Memory and Recall of 500 Images as Attractors of an Autoencoder.** We train a fully connected autoencoder (architecture detailed in Appendix B) on 500 black and white CIFAR10 training images until the training loss is less than $10^{-8}$. We find that all 500 examples are stored as attractors since the top eigenvalue of the Jacobian for all of these examples is less than 1. Lastly, upon iterating 10,000 black and white examples from CIFAR10 and 10,000 examples of Gaussian random noise, we are unable to find any attractors outside the training set.

**Sequence Encoding.** Remarkably, this phenomenon extends from encoding single instances (such as images) to sequences. Instead of autoencoding, where the loss is the reconstruction loss on single images, we train $f$ such that $f(x^{(i)}) \approx f(x^{((i+1) \mod n)})$. In Figure 1b we trained a network to encode 389 frames of size $128 \times 128$ from the Disney film "Steamboat Willie" by mapping frame $i$ to frame $i + 1 \mod 389$. As is shown in the attached video and in Figure 1b, inputting random noise to the trained network $f$ and iterating the network yields the original video. In fact, after a few iterations from a random image, one of the frames of the original video is recovered to numerical precision, and since the network is trained to map $i$ to frame $i + 1 \mod 389$, once one of the frames of the original video is recovered, the remaining frames are all recovered. Interestingly, we show in Section 4 that encoding a sequence of instances in this manner appears to provide a more efficient mechanism for storage and retrieval than autoencoding each instance in the sequence separately.

We can similarly apply the sequence encoding objective to audio. Given an 8 second sample audio, we trained a fully connected network to map each second $i$ of frequencies (represented as a 22k dimensional vector) to second $i + 1 \mod 8$. As demonstrated in the attached audio file, performing iteration starting from random noise yields the entire audio sequence. These two examples demonstrate that both long sequences as well as high-dimensional sequences can be stored and recalled using fully connected networks. In Appendix A, we describe these examples in further detail, and also provide an example of a recurrent network memorizing sequences of text.

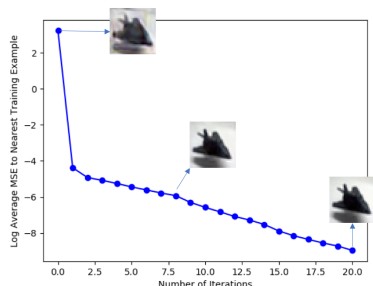
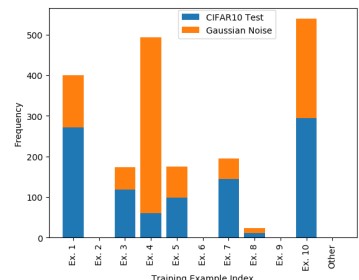

(a) Average log(MSE) between 500 Gaussian Noise inputs iterated 20 times and the closest training example.

(b) Identifying attractors from iterating 1000 test examples from CIFAR10 and 1000 examples of Gaussian noise.

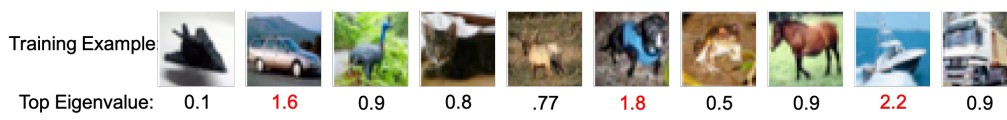

(c) Top eigenvalue of Jacobian at each training image.

Figure 2: Fully connected autoencoder trained on 10 images from CIFAR10.

In the following sections, we will demonstrate that a variety of network architectures can be used to implement associative memory when in the overparameterized setting. In Section 2, we provide some background on dynamical systems and describe how to quantify the number of attractors or limit cycles of autoencoders and sequence encoders. In Section 3, we study the impact of optimizer, nonlinearity, and initialization on the number of training examples stored as attractors for a fixed fully connected network topology. In this section, we also show that convolutional autoencoders, similar to fully connected autoencoders, can store training examples as attractors. We then conclude, in Section 4, with a study of the amount of overparameterization (i.e., depth and width) needed for autoencoders to store individual examples as attractors as compared to the amount needed for sequence encoders to store the same data as limit cycles. Interestingly, we show that sequences can be stored as limit cycles using smaller networks. Details on all architectures and hyperparameters used are provided in Appendix B.

## 2 DYNAMICAL SYSTEMS PERSPECTIVE

After training a fully connected autoencoder $f$ on 10 examples from CIFAR10, we observed in Figure 1a that inputting a random image $x$ to the autoencoder yields an output $f(x)$ that is close* to one of the training examples. Since autoencoders are maps from the feature space to itself, the map can be iterated through $k$ compositions $f^k(x) = f(\ldots(f(x)))$. Figure 2a shows that iteration of the map started at a random image converges to a training example. More precisely, Figure 2a shows that when iterating the convolutional autoencoder on 500 samples of standard Gaussian noise, the distance between the iterated image and the nearest training example decreases monotonically and the reconstruction becomes increasingly accurate. To better understand when this iteration converges to a training image, we take a dynamical systems perspective.

Formally, we let $f : \mathbb{R}^d \to \mathbb{R}^d$ denote the function learned by an autoencoder trained on a dataset $X = \{x^{(i)}\}_{i=1}^n \subset \mathbb{R}^d$. We now consider the sequence $\{f^k(x)\}_{k \in \mathbb{N}}$ where $f^k$ denotes $k$ compositions of $f$ and $x \in \mathbb{R}^d$. When $f(x) = x$, then the sequence $\{f^k(x)\}$ trivially converges to $x$, and we refer to such points $x \in \mathbb{R}^d$ as *fixed points* of the function $f$. Since overparameterized autoencoders interpolate the training data, it holds that $f(x^{(i)}) \approx x^{(i)}$ for each training example $x^{(i)} \in X$ and hence all training examples are fixed points of $f$.[†] In the following, we formally define what it means for a fixed point to be an *attractor* and provide a sufficient condition for this property.

---

*As measured by mean-squared error (MSE).

[†]To ensure $f(x^{(i)}) \approx x^{(i)}$, it is essential that we interpolate to numerical precision i.e. training loss less than $10^{-8}$ when possible.

**Definition.** *A fixed point $x^* \in \mathbb{R}^d$ is an **attractor** if there exists an open neighborhood of $x^*$, such that for any $x$ in that neighborhood the sequence $\{f^k(x)\}_{k \in \mathbb{N}}$ converges to $x^*$ as $k \to \infty$. The set $S$ of all such points is called the **basin of attraction** of $x^*$.*

**Proposition 1.** *A fixed point $x^* \in \mathbb{R}^d$ is an attractor of a differentiable map $f$ if all eigenvalues of the Jacobian of $f$ at $x^*$ are strictly less than 1 in absolute value. If any of the eigenvalues are greater than 1, $x^*$ cannot be an attractor.*

Proposition 1 is a well-known condition in the theory of dynamical systems and follows from the Banach Fixed Point theorem (Rudin, 1964). The condition intuitively means that the function $f$ is "flatter" around an attractor $x^*$.

Returning to overparameterized autoencoders, where each training example is a fixed point, computing the top eigenvalue (in absolute value) of the Jacobian for each training example can be used to determine whether the example is an attractor. Figure 2c lists the maximal eigenvalue of the Jacobian for each training example in this application; 7 out of the 10 training examples are attractors and can hence be recovered through iteration.

To investigate whether there are additional attractors outside of the training examples, we sample random images and iterate the map until convergence. More precisely, we say that iterative application of the autoencoder has converged for an input $x$ if for some $k$, $\|f^k(x) - f^{k+1}(x)\|_2 < 10^{-8}$. We then compare the output $f^k(x)$ to each of the training examples and say that it has converged to one of the training examples if the MSE to the nearest training example is less than $10^{-2}$. In Figure 2b, we iterated 1000 test examples from CIFAR10 and 1000 examples of standard Gaussian noise until convergence. The histogram indicates the number of input patterns whose iterations converged to each training example. Note that no instances converged to the training examples 2, 6 and 9, which is consistent with the fact that these training examples are not attractors (the maximal eigenvalue of the Jacobian is larger than 1). Interestingly, every one of the 2000 instances converged to a training example, i.e., no attractors were found outside the training set. In addition, while it is conceivable that this iterative map may diverge for some inputs $x$, empirically the map converged in all instances.

We end this section by considering the equivalent of an attractor for sequence encoding, namely *discrete limit cycles*. Since each training example maps to the next example in the sequence, none of the training examples are fixed points after training.

**Proposition 2.** *Let $f : \mathbb{R}^d \to \mathbb{R}^d$ be trained to interpolate the training sequence $x^{(1)}, \ldots, x^{(n)}$, i.e., $f(x^{(i)}) = f(x^{((i+1) \mod n)})$. Then the training sequence $\{x^{(i)}\}_{i=1}^n$ forms a discrete limit cycle if the largest eigenvalue of the Jacobian of $f^n(x^{(i)})$ is (in absolute value) less than 1 for any $i$.*

This follows directly from Proposition 1 by considering the map $f^n$, since $x^{(i)} = f^n(x^{(i)})$. To determine other limit cycles of length $n$, one can check whether iterating the sequence encoder from a random point yields a convergent sequence of period $n$.

## 3  IMPACT OF ARCHITECTURE AND OPTIMIZER

We start this section by analyzing how different optimizers, initializations and nonlinearities influence the number of training examples that are stored as attractors for fully connected autoencoders; we discuss convolutional autoencoders at the end of this section. For fully connected networks, we base our analysis of storage and retrieval on a dataset of 100 black and white images from CIFAR10[‡].

### 3.1  MEMORIZATION AND RECALL ACROSS OPTIMIZERS AND NONLINEARITIES

We begin by studying how changing the optimizer and nonlinearity affects the number of training examples stored as attractors. In Figure 3, we provide the number of training examples that become attractors when using popular training algorithms including gradient descent (GD), GD with momentum, GD with momentum and weight decay, RMSprop, and Adam (Ch. 8 of Goodfellow et al. (2016)) with popular nonlinearities including ReLU, Leaky ReLU, SELU, cosid (i.e. $\cos x - x$), Swish (Xu et al., 2015; Ramachandran et al., 2017; Eger et al., 2018) . We used a learning rate of

---

[‡]We chose to use black and white images in order to make convergence to MSE $\leq 10^{-8}$ and the computation of the Jacobian comptationally feasible (for each of the 100 examples the top eigenvalue among all 1024 eigenvalues of the Jacobian needs to be computed).

| Act.
Opt. | ReLU | Leaky
ReLU | SELU | Swish | $\cos x - x$ | $x + \frac{\sin 10x}{5}$ |
|---|---|---|---|---|---|---|
| GD | 28/100 | 34/100 | 10/100 | NA* | 5/100 | 19/100 |
| GD +
Momentum | 14/100 | 23/100 | 10/100 | NA* | 2/100 | 21/100 |
| GD +
Momentum
+ Weight
Decay | NA* | NA* | 18/100 | NA* | 22/100 | NA* |
| RMSprop | 97/100 | 98/100 | 100/100 | 49/100 | 100/100 | 100/100 |
| Adam | 38/100 | 53/100 | 30/100 | 14/100 | 100/100 | 100/100 |

Figure 3: Impact of optimizer and nonlinearity on number of training examples stored as attractors. In all experiments, we used a fully connected network with 11 hidden layers, 256 hidden units per layer, and default PyTorch initialization. (∗) NA indicates that the training error did not decrease below $10^{-5}$ in 1,000,000 epochs.

$10^{-1}$ for GD methods and $10^{-4}$ for RMSprop and Adam. We used a momentum value of 0.009 and weight decay of 0.0001. Overall, we observed that using adaptive methods with a learning rate per parameter can significantly alter the number of training examples stored as attractors. Notably, we found that RMSprop consistently led to nearly all 100 examples being stored as attractors even across different nonlinearities. We note that methods using gradient descent were very slow to converge; in particular, the loss often did not reduce to less than $10^{-5}$ in over a million epochs (denoted by NA in Figure 3).[§]

Interestingly, when analyzing whether there are any attractors other than the training examples, we observed that the GD methods tended to have a larger number of attractors outside the training set than adaptive methods. Figures 11a and 11b in Appendix C show the distribution of attractors found through iteration for SELU networks trained using GD with momenutm and weight decay and RMSprop. Figure 11c in Appendix C provides examples of attractors outside the training set for the network trained with GD + momentum + weight decay.

With respect to nonlinearities, Figure 3 shows that trigonometric nonlinearities allow for all 100 training examples to be stored as attractors when using RMSprop or Adam. Figure 12 in Appendix C provides the distributions of attractors found through iteration for a subset of these settings. Overall, these plots indicate that the trignometric nonlinearities have fewer attractors outside the training set than the piecewise activation functions (ReLU, Leaky ReLU, SELU). Notably, we did not find any attractors outside the training set using the cosid nonlinearity and the Adam optimizer.

## 3.2 MEMORIZATION AND RECALL ACROSS INITIALIZATIONS

In the following, we study the impact of initialization on the number of training examples that are stored as attractors of the learned map. For all experiments in this section, we used a fully connected network with 11 hidden layers and 256 hidden units per layer that is trained using Adam until MSE $\leq 10^{-8}$. Thus far, we have been using the default initialization provided from PyTorch (Paszke et al., 2017).[¶] Under this initialization scheme, each layer's weights are drawn i.i.d. from a uniform distribution $U[-a, a]$ with $a = 1/\sqrt{h}$, where $h$ is the number of hidden units in that layer. Since our architecture has 256 hidden units per layer, $a = 0.0625$ for all but the last layer by default. We note that since we use 256 hidden units in each layer, with appropriate constants, this initialization is the same as the Xavier or Kaiming uniform initializers for almost all layers (Glorot & Bengio, 2010; He et al., 2015).

In Figure 4, we demonstrate the tradeoff between nonlinearity and uniform initialization scheme. We observe that as we increase $a$, the number of training examples stored as attractors mostly decreases. In fact, for large values of $a$ such as $a = .15$, not only are fewer training examples attractors for these activation functions, but iteration starting from random examples often diverges. In Figure 13

---

[§]Unlike our other experiments, we here choose an MSE threshold of $10^{-5}$, as almost none of non-adaptive optimizers could reduce the loss below $10^{-8}$ in under $1,000,000$ iterations.

[¶]We used PyTorch version 0.4; note that the default initialization scheme may be different in later versions.

| Act. Init. | ReLU | Leaky ReLU | SELU | Swish | $\cos x - x$ | $x + \frac{\sin 10x}{5}$ |
|---|---|---|---|---|---|---|
| $U[-0.01, 0.01]$ | 62/100 | 78/100 | 78/100 | 16/100 | 26/100 | 93/100 |
| $U[-0.02, 0.02]$ | 43/100 | 65/100 | 71/100 | 20/100 | 31/100 | 70/100 |
| $U[-0.05, 0.05]$ | 55/100 | 55/100 | 29/100 | 32/100 | 100/100 | 89/100 |
| $U[-0.1, 0.1]$ | 36/100 | 43/100 | 13/100 | 30/100 | 100/100 | NA* |
| $U[-0.15, 0.15]$ | 34/100 | 38/100 | 13/100 | 6/100 | 100/100 | NA* |

Figure 4: Impact of initialization on number of training examples stored as attractors. In all experiments, we used a fully connected network with 11 hidden layers and 256 hidden units per layer trained using the Adam optimizer (lr=$10^{-4}$). (∗) NA indicates that the training error did not decrease below $10^{-8}$ in 1,000,000 epochs.

of Appendix C, we again provide the distribution of attractors found through iteration for a subset of these settings.

### 3.3 MEMORIZATION AND RECALL IN CONVOLUTIONAL AUTOENCODERS

Thus far, we have provided several examples showing that fully connected autoencoders can store training examples as attractors. We now demonstrate that convolutional autoencoders can also exhibit this behavior. For this, we trained a U-Net convolutional autoencoder (detailed in Appendix B) on 10 CIFAR10 training examples (Ronneberger et al., 2015; Ulyanov et al., 2017). We performed this analysis only on 10 training images due to the high computational cost of training a U-Net convolutional autoencoder to MSE $\leq 10^{-8}$ (roughly 9 hours on 10 training images) and computing the top eigenvalue for each of the 10 examples ($\sim 1$ hour). For this reason, we also focused the analysis in the previous sections on fully connected networks.

Figure 5a shows that, as for fully connected networks, inputting test examples from CIFAR10 or Gaussian noise into the trained U-Net convolutional autoencoder leads to images that are visually indistinguishable from the training examples. In fact, since the eigenvalues of the Jacobian are less than 1 for each training example, all 10 training examples are attractors. In addition, Figure 5b shows that iterating the map starting in 1000 examples of CIFAR10 test examples or 1000 examples of Gaussian noise did not lead to the discovery of any attractors outside the training set.

## 4 MEMORIZATION AND RECALL OF SEQUENCES

We have thus far considered the impact of optimizer, nonlinearity, and initialization on the number of training examples stored as attractors in autoencoders. In this section, we first study how changing

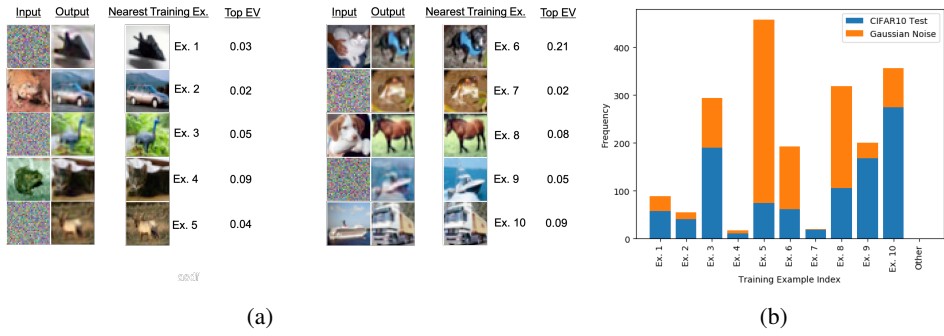

(a)          (b)

Figure 5: (a) U-Net convolutional autoencoder trained on 10 images from CIFAR10. When fed Gaussian noise or test images from CIFAR10 the model outputs images that are visually indistinguishable from training images. All training images are attractors (EV is eigenvalue). (b) Identifying attractors from iterating the trained autoencoder starting in 1000 test examples from CIFAR10 and 1000 examples of Gaussian noise. No attractors are found outside the training set.

| Depth \ Width | 128 | 256 | 512 |
|---|---|---|---|
| 1 | 0 | 0 | 0 |
| 6 | 0 | 4 | 12 |
| 11 | 2 | 8 | 24 |
| 16 | 22 | 38 | 49 |
| 21 | 56 | 68 | 75 |
| 26 | 83 | 86 | 94 |
| 31 | 92 | 90 | 99 |

Figure 6: Impact of width and depth on number of MNIST examples (out of 100) stored as attractors. The networks used have SELU activations, default initialization and are trained using Adam until MSE $\leq 10^{-8}$.

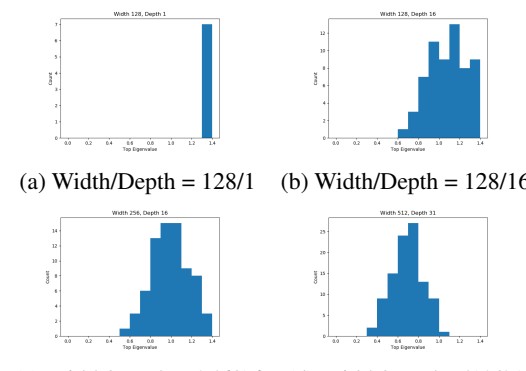

(a) Width/Depth = 128/1    (b) Width/Depth = 128/16

(c) Width/Depth = 256/16    (d) Width/Depth=512/31

Figure 7: Top eigenvalues of Jacobian for all 100 training examples.

the width and depth of a fully connected autoencoder impacts the number of training examples stored as attractors. In particular, we observe that the number of training examples that are stored as attractors increases as depth and width increase. We then study how sequence encoders can store the same data as limit cycles. Interestingly, we find that storing the data as a sequence in a limit cycle requires less width and depth than storing the images individually. In order to improve computational efficiency when training to a loss of $10^{-8}$ using Adam and computing the eigenvalues of the Jacobian, in this section, we use 100 examples from MNIST as training data. In all of our experiments, we use fully connected SELU networks and fix the random seed prior to training for reproducibility.

In Figure 6, we demonstrate how increasing depth and width lead to the storage of almost all training examples as attractors. Note that a minimum width of 100 is needed to allow for interpolation. Interestingly, when there is sufficient depth for storing at least 1 training example as an attractor, we observe that increasing width leads to the storage of more training examples as attractors. Figure 7 shows that the top eigenvalues of the Jacobian for all 100 examples decrease when increasing depth and width.

Instead of training autoencoders to store individual images as attractors, we can instead encode MNIST digits as sequences and then use sequence encoding to store the sequences as limit cycles. We consider the following example of encoding two 10-digit sequences from MNIST: one counting upwards from digit 0 to 9 and the other counting down from digit 9 to digit 0. The training data is presented in Figure 8. Using a fully connected network with a depth of 31 layers, 256 hidden units per layer, and SELU activations trained using the Adam optimizer until MSE $\leq 10^{-8}$, we find that iterating Gaussian inputs through the encoder leads to recovery of both training sequences. This is quite surprising, since even though digits from the two sequences are very similar, iterating the map converges precisely to one of the two sequences (i.e., there are no jumps between a digit of one sequence and the other). Applying Proposition 2, we find that the maximal eigenvalues of the

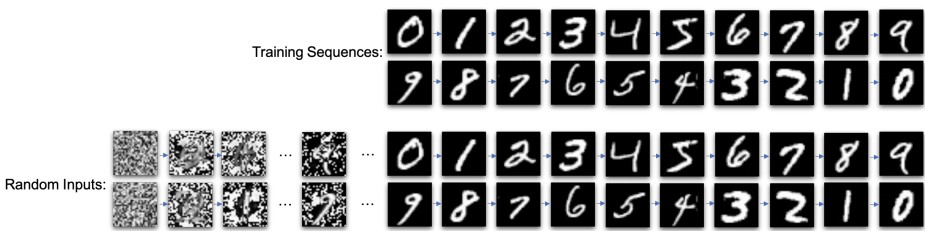

Figure 8: 2 MNIST digit training sequences of length 10 stored as limit cycles of an encoder. Iteration from random noise converges to each of the sequences. Our network has 256 hidden units per layer, 31 hidden layers, SELU activations, default initialization, and is trained using the Adam optimizer until MSE $\leq 10^{-8}$.

| Depth \ Width | 128 | 256 | 512 |
|---|---|---|---|
| 1 | 0 | 0 | 0 |
| 6 | 0 | 3 | 10 |
| 11 | 0 | 5 | 10 |
| 16 | 0 | 9 | 10 |
| 21 | 6 | 9 | 10 |
| 26 | 8 | 10 | 10 |
| 31 | 10 | 10 | 10 |

(a) Impact of width and depth on storing 10 MNIST sequences of length 10 as limit cycles.

| Depth \ Width | 128 | 256 | 512 |
|---|---|---|---|
| 1 | ✗ | ✗ | ✓ |
| 6 | ✗ | ✓ | ✓ |
| 11 | ✗ | ✓ | ✓ |
| 16 | ✗ | ✓ | ✓ |
| 21 | ✗ | ✓ | ✓ |
| 26 | ✗ | ✓ | ✓ |
| 31 | ✓ | ✓ | ✓ |

(b) Impact of width and depth on storing 1 MNIST sequence of length 100 as a limit cycle.

Figure 9: Storing 100 MNIST examples through sequences. The networks used have SELU activations, default initialization and are trained using Adam until MSE $\leq 10^{-8}$.

Jacobian of the trained encoder composed 10 times is 0.0034 and 0.0033 for the images from the first and second sequence, respectively. Hence, we conclude that these two sequences are indeed limit cycles of the trained encoder.

In Figure 6, we saw that none of the analyzed fully connected networks stored all 100 training examples as attractors. Instead of using our architecture as an autoencoder, we can instead partition the 100 examples into sequences and train the network to encode each of the sequences. In Figure 9 we investigate whether it is easier to store shorter or longer sequences as limit cycles. In particular, we begin by grouping the 100 training images into 10 sequences of length 10. In Figure 9a, we study how width and depth affects the number of sequences that become limit cycles of the network. Interestingly, while none of the analyzed architectures were able to store the 100 training images as attractors in the autoencoding setting, already a small network of depth 6 and width 512 can store all training examples as 10 limit cycles of length 10 when encoding the training images as sequences.

Given this intriguing observation, we end with the following experiment: Instead of encoding 10 sequences of length 10, we encode all 100 training examples into a single sequence of length 100. Consistent with the above findings, Figure 9b shows that an even smaller network can be used to store all 100 training images in this manner. Interestingly, the sequence of 100 images is a limit cycle of a network with only 1 hidden layer and 512 hidden units.

## 5 CONCLUSION

In this work, we demonstrated experimentally that overparameterized autoencoders and sequence encoders can be used to implement associative memory by storing training examples as attractors or limit cycles. We showed that this remarkable phenomenon is pervasive in a range of common architectures using the standard fully connected or convolutional layers, across several popular optimizers, nonlinearities, and initializations schemes. In fact, by utilizing the architecture and optimizer that provided the most attractors empirically, we were able to train a network that could store and recover 500 real-valued images with no apparent spurious attractors. We further showed that in fully connected architectures, increasing the depth or width could lead to a greater number of training examples or sequences being stored as attractors or limit cycles. Lastly, we demonstrated that networks are able to store and retrieve more training examples by training to encode sequences instead of autoencoding individual instances.

While our paper concentrates on the question of implementing associative memory, we employ essentially the same training procedures and neural net architectures as those used in standard supervised learning. Thus the surprising fact of the existence and, indeed, the ubiquity of attractors in these maps may shed light on the important question of inductive biases of interpolating neural networks (Belkin et al., 2019).

Finally, another avenue for future exploration is the connection of autoencoding and sequence encoding to neural net memory mechanisms in biological systems. In fact, this question had been one of the main motivations for the original work on Hopfield networks (Hopfield, 1982).

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

## APPENDIX A: DATASET DESCRIPTIONS AND MEMORIZATION IN RNNs

**Disney "Steamboat Willie" Video Clip.** In order to demonstrate that fully connected networks can memorize long sequences, we captured 389 frames of size $128 \times 128$ from the Disney movie "Steamboat Wille." The architecture used to memorize and recall the video is provided in Appendix B. We have attached a full video titled "disney_steamboat_willie_recovered_from_noise.mp4" displaying that iteration from random noise leads to recovery of the entire video. Note that iteration from random noise converges so fast that it is difficult to see that we began the attached video from a random noise image. This is more clearly displayed in Figure 1b.

**Donald Trump Audio Sample.** In order to demonstrate that fully connected networks can memorize high dimensional sequences, we captured an audio clip of an 8 second recording from the Donald Trump talking pen. Each second of the audio is roughly $22,000$ frequencies, and we trained a fully connected network to map from the frequencies in second $i$ to second $i + 1 \mod 8$. We have attached an audio sample titled "trump_quote_recovered_from_noise.mp3" demonstrating that iteration from random noise leads to recovery of the entire quote. The full architecture used is provided in Appendix B.

**Text Memorization.** In order to demonstrate that recurrent neural networks (RNNs) can also memorize and recall sequences, we trained a vanilla RNN (whose architecture is detailed in Appendix B) to encode the following sentence from our introduction: "Hopfield networks are able to store binary training patterns as attractive fixed points." When training the RNN, we encode each word using 1-hot representation i.e., since there are 13 words in the sentence, we represent each word with a vector of size 13 and place a "1" in the index corresponding to the word. We train such that each word is mapped to the next modulo 13 using the Cross Entropy Loss (as is done in practice). Unlike the other settings, RNNs are used to generate new sequences after training by sampling a new word from the vector output given a previous word[||]. Under our architecture, we find that repeatedly choosing the highest probability word given the previous word consistently outputs the entire training sentence regardless of the number of times this sampling process is repeated. Note that if the number of hidden units per layer is halved in our architecture, repeating this sampling process will only output part of the training sequence. Namely, only the sequence "Hopfield patterns as attractive fixed points" will ever be generated.

## APPENDIX B: ARCHITECTURES

1. For Figure 1a, we used a fully connected autoencoder with 31 hidden layers, 256 hidden units per layer, SELU activation, and default PyTorch initialization (random seed 2). We trained using the Adam optimizer until the training loss was less than $10^{-8}$.

2. For Figure 1b, we used a fully connected architecture with 16 hidden layers, 1024 hidden units per layer, SELU activation, and default PyTorch initialization (random seed 2). We trained using the Adam optimizer until the training loss was less than $10^{-7}$.

3. For the audio example in Section 1, we used a network with 36 hidden layers, 15 hidden units per layer, SELU activation, and default PyTorch initialization (random seed 2). We trained using the Adam optimizer until the training loss was less than $10^{-12}$ (we need high precision reconstructions to avoid noisy artifacts in the audio file).

4. For the RNN example in Appendix A, we used two fully connected networks with 31 hidden layers and 128 hidden units per layer such that one constructs the next hidden state and the other constructs the next output. We used SELU activations throughout and trained using Adam with learning rate $10^{-4}$ to minimize the Cross Entropy Loss.

5. To encode 500 black and white CIFAR10 images as attractors, we used a fully connected architecture with 11 hidden layers, 1024 hidden units per layer, cosid activation, and default PyTorch initialization (random seed 2). We trained using the Adam optimizer until the training loss was less than $10^{-8}$.

6. For the convolutional autoencoder used for Figure 5a, we display the architecture in Figure 10. We again used the default PyTorch initialization (random seed 2) and trained using Adam until the MSE was less than $10^{-8}$. All of our filters have a kernel size of 3.

---

[||]This process is usually started from inputting the all zero vector.

| x5 | | x5 | | x2 | |
|---|---|---|---|---|---|
| Conv 256 Stride=2 Leaky RELU | Conv 256 Stride=1 Leaky RELU | Bilinear Upsampling | Conv 256 Stride=1 Leaky RELU | Conv 3 Stride=1 Leaky RELU |

Figure 10: Convolutional autoencoder architecture used in our experiments.

7. We used a random seed of 2 for all experiments in Figures 3, 4, 6, 8, 9. As these experiments were computationally expensive, we could not run all of them across several random seeds. However, for a few of the entries in each table, we note that other random seeds gave similar results.

## APPENDIX C: DISTRIBUTION OF ATTRACTORS DISCOVERED THROUGH ITERATION

In this section, we provide a series of histograms indicating the attractors found by iterating test examples from black and white versions of CIFAR10 images and from Gaussian random noise. The histograms also indicate an estimate for the size of the basin of attraction for each attractor.

In Figure 11, we provide histograms for two different optimization methods indicating the distribution of attractors found by iterating 1000 test examples from CIFAR10 and 1000 examples of Gaussian noise. To compare different optimization methods, we used an architecture with SELU nonlinearity and the default PyTorch initializer. The histograms demonstrate that adapative optimizers such as RMSprop lead to the storage of more training examples as attractors. Furthermore, when using RMSProp we did not observe any attractors outside the training set. On the other hand, we identified attractors outside the training set when using GD + momentum + weight decay, and examples of these attractors are presented in Figure 11c.

In Figure 12, we provide histograms for four different nonlinearities indicating the distribution of attractors found by iterating 1000 test examples from CIFAR10 and 1000 examples of Gaussian noise. To compare nonlinearities, we used an architecture that was trained using the Adam optimizer and that was initialized using the default PyTorch initializer. The histograms indicate that trignometric nonlinearities tend to store more training examples as attractors while also having fewer spurious attractive states. Interestingly, the cosid nonlinearity stored all 100 training examples as attractors and had 0 other observed attractors outside the training set.

In Figure 13, we provide histograms for four different initializations indicating the distribution of attractors found by iterating 1000 test examples from CIFAR10 and 1000 examples of Gaussian noise. To compare different initializations, we selected an architecture that was trained using the Adam optimizer and used the SELU nonlinearity. The histograms indicate that smaller variance

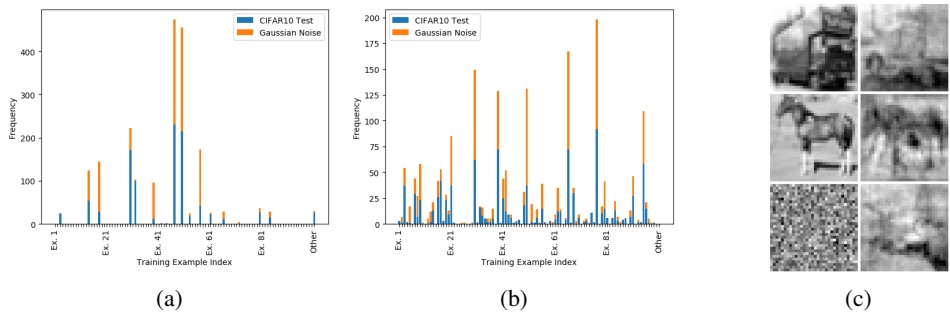

(a)          (b)          (c)

Figure 11: Identifying attractors from random inputs across optimizers for the SELU nonlinearity. (a) SELU network trained using GD + momentum + weight decay; 6 observed attractors outside training set. (b) SELU network trained using RMSprop; 0 observed attractors outside training set. (c) Examples of attractors outside training set for SELU network trained using GD + momentum + weight decay.

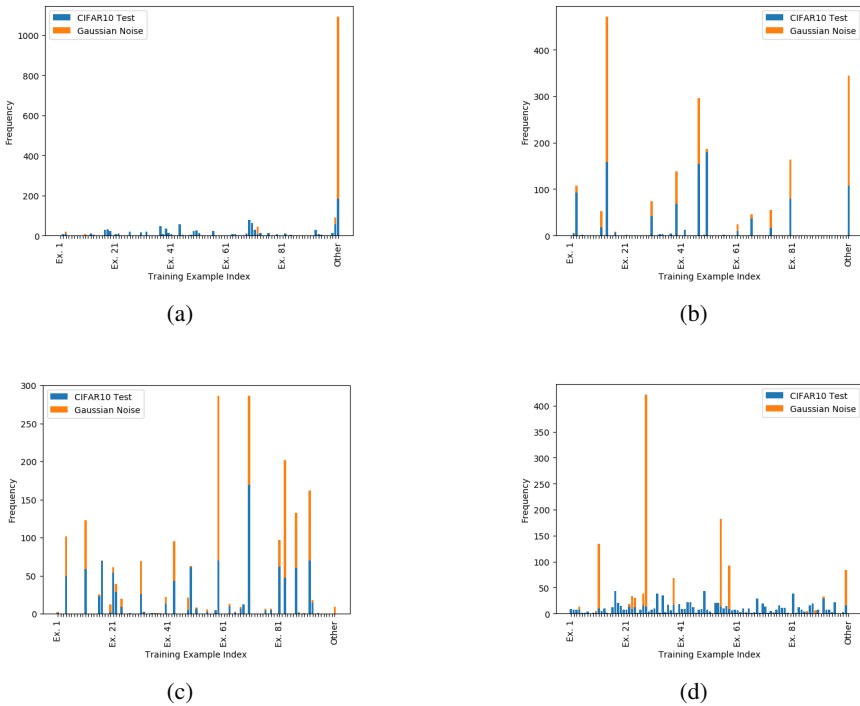

Figure 12: Identifying attractors from random inputs across nonlinearities.(a) Leaky RELU network trained using Adam; 59 observed attractors outside trainset. (b) SELU network trained using Adam; 17 observed attractors outside trainset. (c) Network with $x + \frac{\sin 10x}{5}$ activations; 9 observed attractors outside trainset. (d) Network with $\cos x - x$ activations; 0 observed attractors outside trainset.

initializations tend to store more training examples as attractors while also having fewer spurious attractive states.

In Figure 14, we provide a histogram indicating the distribution of attractors found by iterating 10,000 test examples from CIFAR10 and 10,000 examples of Gaussian noise for a fully connected network trained to store 500 examples. We selected an architecture that was trained using the Adam optimizer, used the cosid nonlinearity, was initialized using the default PyTorch initialization, and that had 11 hidden layers with 1024 hidden units per layer. Notably, we did not observe any other attractor states from this network, even though we iterated 20,000 examples.

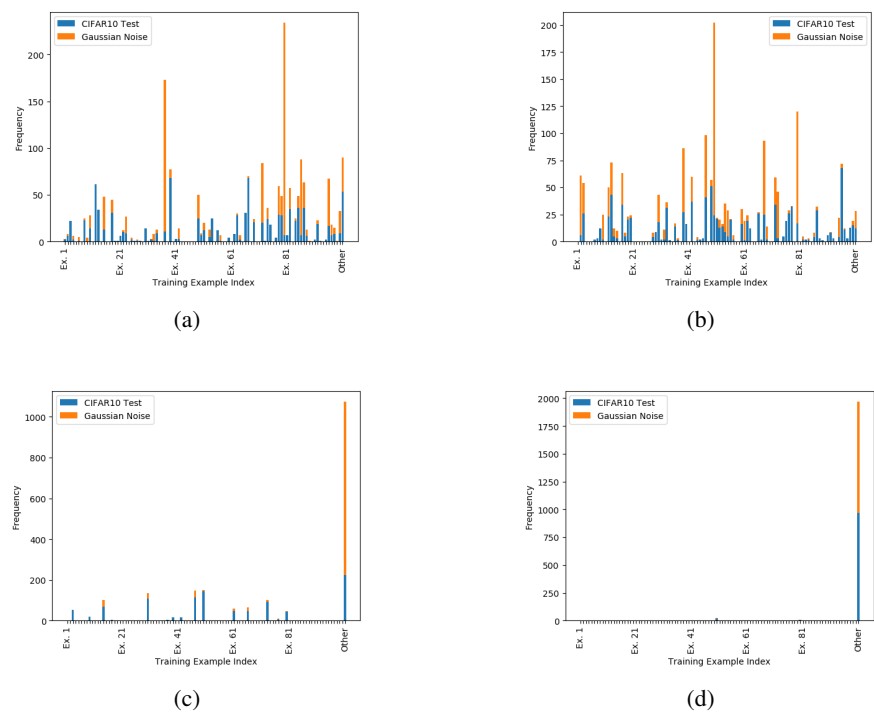

Figure 13: Identifying attractors from random inputs across initializations. (a) SELU network trained with Adam with initialization $U[-0.01, 0.01]$; 5 observed attractors outside the training set. (b) SELU network trained with Adam with initialization $U[-0.02, 0.02]$ ; 8 observed attractors outside the training set. (c) SELU network trained with Adam with initialization $U[-0.05, 0.05]$, 12 observed attractors outside the training set. (d) SELU network trained with Adam with initialization $U[-0.1, 0.1]$; almost no inputs converged to training examples within 1000 iterations.

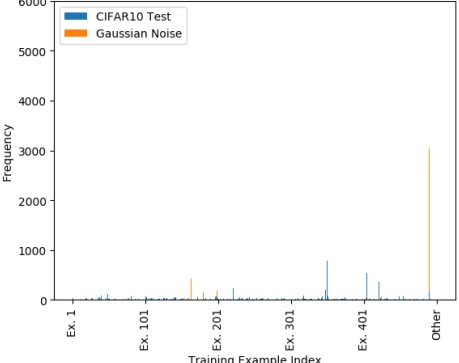

Figure 14: Iterating 10,000 test examples and 10,000 examples of random noise does not lead to any observable attractors outside the training set.

