# OpenReview forum: "Overparameterized Neural Networks Can Implement Associative Memory"
_ICLR.cc/2020/Conference — Reject_

### Official Review · AnonReviewer2 · 2019-10-19
**Official Blind Review #2**

**Rating:** 3

**Review:**

The paper studies a phenomenon of unusual memorisation in deep overparametrized neural networks.
Authors observe that, if an auto-encoder overfits to machine precision on a number of images, they can be reliably decoded from random noise and that it is even possible to memorise this way a sequence of images.
Essentially, images from such a training set become attractors for the mapping defined by the auto-encoder.
The impact of network size, nonlinearity and initialization is studied and, quite surprisingly, very unusual trigonometric non-linearities performed the best.

I find the studied phenomenon rather interesting and the analysis well-performed, but I am not sure how practically important is this work.
First, I would argue that to call the overfit auto-encoder a function associative memory, it must be able to retrieve stored images not just from random noise, but from a somehow distorted or partially known version.
Otherwise we are just left with a ridiculously large network that can only recall a handful of images we could store in the raw format using much less numbers.
Second, training until convergence takes prohibitively long time.

I would be also interested to at least an interesting discussion, if not an answer, to the question of why and how exactly trained images become attractors.

In terms of novelty, it feels like Zhang et al, 2019 already studied a very similar phenomenon and the submitted paper does not add much to understanding of memorisation in neural networks. However, memorization of sequences was indeed a surprise.

Overall, I do not have a strong opinion on rejecting the paper, it just feels like more work in this direction will make the paper significantly better.

**Experience Assessment:**

I have published one or two papers in this area.

**Review Assessment: Checking Correctness Of Derivations And Theory:**

I assessed the sensibility of the derivations and theory.

**Review Assessment: Checking Correctness Of Experiments:**

I carefully checked the experiments.

**Review Assessment: Thoroughness In Paper Reading:**

I read the paper at least twice and used my best judgement in assessing the paper.

---

> ### Author Response · Authors · 2019-11-07
> **Response to Review #2**
>
> We thank the reviewer for the comments and  emphasizing that the phenomenon we identify is interesting and that our analysis is well performed.
>
> Regarding the comparison to Zhang et al, 2019:  We would like to point out that the first arXiv version of our paper precedes the first version of Zhang, et al. Just like our paper, Zhang, et al has only been presented in a workshop, which does not count as a refereed publication.  Furthermore, a version of Zhang, et al is concurrently under review in this same venue.   Therefore, we strongly feel that our paper should be reviewed on its own merits and not compared against Zhang, et al.
>
> In terms of the importance of this result, our method is the first for training a model to store and recover high dimensional inputs up to numerical precision. Moreover, there is considerable interest in understanding the similarities and differences in artificial neural networks and biological neural networks. Our results point to a biologically plausible mechanism for memory retrieval (while the biological plausibility of the training process still remains an open question). Namely, we show that iterating a trained autoencoder allows retrieving stored images.  Our results also suggest an interesting hypothesis for biological neural networks, which we are currently following up on with neuroscientists. We demonstrated that it is “easier” for an artificial neural network to store sequences of images instead of individual images, or more precisely, smaller networks can be used to store sequences of images as compared to the same number of single images. Similar phenomena may be observed in biological neural networks.
>
> Moreover, a question of considerable interest in machine learning is the identification of the inductive bias of neural networks. In the overparameterized setting, a neural network can achieve zero training error. There are many different functions that can achieve zero training error. What are the functions learned by a neural network, i.e. what is its inductive bias? Our study identifies a novel form of inductive bias of deep networks that persists across different architectures: deeper networks tend to store training examples as attractors. This means that deep networks learn functions that are contractive at the training examples, a form of self-regularization.
>
> By the definition of an attractor, iterating the network on points within an open set around an attractor will converge to the attractor.  Hence, the model does represent associative memory as small perturbations to an attractor (i.e. a point within this open set) will converge to the attractor upon iterating the network.  Importantly, this condition is a mathematical guarantee for associative memory.  Hence, the networks learned do implement associative memory mechanisms with mathematically verifiable conditions on which training examples are attractors.  We will add some experiments to highlight this in the revision.

---

> > ### Comment · AnonReviewer2 · 2019-11-15
> > **Response to response**
> >
> > Thank you for your reply.
> >
> > I deliberately did not search for your paper on arxiv to not hinder the double blind review process. I agree that the paper of Zhang et al probably should not diminish novelty of your results. However I find it a strange practice to ignore the existence of these results and not citing it since you are aware of them.
> >
> > I remain the same in my understanding of what a practically useful associative memory is. While you can prove that the training images are attractors, it is important either to provide some theoretical insight or empirical results on whether other spurious attractors exist in the system or what are the basins of these "genuine" attractors. To me it is important and definitely possible to conduct a set of at least basic experiments to further investigate properties of the proposed memory system.

---

> > > ### Author Response · Authors · 2019-11-15
> > > **Rebuttal**
> > >
> > > We would appreciate if the reviewer could actually read our paper. We cite Zhang et al, 2019 in the first paragraph of the introduction. We do not cite our own paper, which precedes Zhang et al, in order to comply with the double blind policy.
> > >
> > > We have done an extensive investigation of these basins of attraction.  We already show an estimate of the basin of attraction for each training example in Figures 2b, 5b, 11, 12, 13, and 14.  We also provide examples of spurious attractors in Figure 11.  Lastly, we also iterated 20,000 images to estimate the basin of attraction for 500 training images in Figure 14.  The  attractors we identify are indeed mathematically genuine as we verify that all eigenvalues of the Jacobian are less than 1 in absolute value.

---

> > > > ### Author Response · Authors · 2019-11-15
> > > > **No Comment Deleted**
> > > >
> > > > Update: Just to clarify,  no comment has been deleted.  We are not sure why our answer now appears after your answer.  In addition, we have also answered to your other comments.

---

> > > ### Comment · AnonReviewer2 · 2019-11-15
> > > **Correction**
> > >
> > > Apologies, I have got confused about the not citing Zhang et al.
> > > The discussion phase is overwhelming for authors, reviewers and combinations of thereof and we all can make mistakes due to the volume of work. You can always correct a reviewer in a better tone than in your deleted comment. I strongly recommend to abstain from such behavior in the future.
> > >
> > > The rest of my review remains the same and I am disappointed that authors decided not to answer to my initial suggestion of a more extensive experimental study.

---

### Official Review · AnonReviewer1 · 2019-10-21
**Official Blind Review #1**

**Rating:** 3

**Review:**

This paper empirically demonstrates that DNNs can be trained to be identity mappings for small quantities of samples. It also demonstrates that for many parameterizations, these identity DNNs also have a small number of attractors, iterative fixed points, and can also learn short circular sequences of examples.

The paper is well written and easy to understand, although the presentation could be improved a bit (see comments). Its contents aren't particularly novel in terms of ideas, but they investigate memorization and attractors much further than previous studies.
In a way, the memorization results are unsurprising. We know that DNNs can memorize perfectly, including sequences, so it is natural that by increasing capacity, at some point they should be able to memorize entire images (in fact this is what Zhang et al. (2019)'s Figure 1 appears to be showing).
The more novel and surprising aspect of this is that DNNs would learn such strong (and so few) attractor basins. The fact that deep autoencoders could have attractors centered on the training points has been postulated before (see [1]), but this work makes a stronger case for it.

A crucial aspect that is missing from this paper in order for me to give in an accept is that there is very little about how this paper positions itself in the current literature. There could be much more discussion about related work, and much more discussion about the impacts of these findings.

I have given this paper a 'weak reject' mark but I think with some work this paper could be of interest to many. To reiterate, I am unable to see anything wrong with this paper, but at the same time I am unable to see how impactful these findings are.


Detailed comments:
- It's interesting that DNNs can implement associative memory, but what is the cost of doing that? Should we be using that in practice? Since there is no sense of how costly the presented experiments are, it is hard to tell.
- Again, these results are interesting, but after some time pondering about it, I can't really convince myself that knowing the results of this paper will be beneficial to future research. That being said, there are many areas of Machine Learning that I am unfamiliar with. It should be part of the paper to familiarize readers with areas where these results could be impactful.
- "the function interpolates the training images" not sure what this means. Interpolation means making a prediction for a point `u` that is "between" two points `x,y` with known values
- "black and white" should be "grayscale" if values are in [0,1]
- Figure 2b is interesting, but I wonder what happens if e.g. a perturbed version of e.g. Example 6 is fed. Presumably since Example 6 is not an attractor (Jacbian with an eigeinvalue > 1), it should converge to another example.
- Figure 2b's caption numbers, which say you use 1k example, to not correspond to numbers earlier in the text, which say you use 10k examples.
- "Since overparameterized autoencoders interpolate the training data", again this is a fairly important assumption and it needs to be defined very clearly, because it could mean many things.
- "it is essential that we interpolate to numerical precision", I don't think you are using the word "interpolate" correctly, do you mean "inference"? "train"?
- Adam citation should be "Adam: A Method for Stochastic Optimization, Diederik P. Kingma, Jimmy Ba", not Goodfellow et al., RMSprop should also have a citation, Hinton et al. 2012
- ReLU citation should be "Rectified linear units improve restricted Boltzmann machines, Nair & Hinton", Leaky ReLU should be Maas et al 2013, SELU should be Klambauer et al. 2017.
- The combination of section 3.1 and Figure 3 doesn't make it clear if models trained with Adam and RMSprop have weight decay or not. Can you clarify?
- "Note that a minimum width of 100 is needed to allow for interpolation." Again I think you mean "learning" rather than "interpolation".
- You say that you trained black and white images, but all the images of CIFAR10 in the figures are colored, including the inputs and outputs. Can you clarify why?
- You might be interested in [2], which is much older work about perceptrons, but still relevant to what is studied here.
- The linked supplemental material gives a 404 for me. I replicated the MNIST Figure 6 experiment. I was unable to replicate exactly your results but they were similar enough. In particular, the activation function choice seems to be critical.

[1] The Potential Energy of an Autoencoder, Hanna Kamyshanska, Roland Memisevic
[2] Basins of Attraction in a Perceptron-like Neural Network, Werner Krauth, Marc Mezard, Jean-Pierre Nadal



**Experience Assessment:**

I have published one or two papers in this area.

**Review Assessment: Checking Correctness Of Derivations And Theory:**

I assessed the sensibility of the derivations and theory.

**Review Assessment: Checking Correctness Of Experiments:**

I carefully checked the experiments.

**Review Assessment: Thoroughness In Paper Reading:**

I read the paper thoroughly.

---

> ### Author Response · Authors · 2019-11-07
> **Response to Review #1**
>
> Thank you for the careful reading of our paper and helpful comments.
>
> Comparison to Zhang et al, 2019.  We would like to point out that the first arXiv version of our paper precedes the first version of Zhang, et al.  Just like our paper, Zhang et al has only been presented in a workshop, which does not count as a refereed publication.  Furthermore, a version of Zhang et al is concurrently under review in this same venue.   Therefore, we strongly feel that our paper should be reviewed on its own merits and not compared against Zhang, et al.
>
> In addition, we would like to clarify the following comment:
> “The fact that deep autoencoders could have attractors centered on the training points has been postulated before (see [1])”.
>
> It is well-known that attractors can be used as a form of memory, e.g., Hopfield networks.  However, the work [1] and to the best of our knowledge other work in this area do not observe the central phenomenon of our submission, the emergence of attractors in overparameterized networks trained using standard optimization methods.
>
> In the following, we provide a point-by-point response to the detailed comments provided by the reviewer.
>
> (1) The computational cost of these methods is an interesting question as a connection to biological memory, but at this point, we concentrate on identifying the phenomenon.
>
> (2) With respect to the impact of our results: (a) A question of considerable interest in machine learning is identifying the inductive bias of neural networks. In the overparameterized setting, a neural network can achieve zero training error. There are many functions that can achieve zero training error. What are the functions learned by a neural network, i.e. what is its inductive bias? Our study identifies a novel form of inductive bias of deep networks that persists across different architectures: deeper networks tend to store training examples as attractors. This means that deep networks learn functions that are contractive at the training examples, a form of self-regularization. (b) There is considerable interest in understanding the similarities and differences in artificial and biological neural networks. Our results provide a biologically plausible mechanism for memory retrieval. Namely, we show that iterating a trained autoencoder allows retrieving stored images. (c) Our results suggest an interesting hypothesis for biological neural networks, which we are currently following up on with neuroscientists. We demonstrated that it is “easier” for an artificial neural network to store sequences of images instead of individual images, or more precisely, smaller networks can be used to store sequences of images as compared to the same number of single images. Similar phenomena may be observed in biological neural networks. We thank the reviewer for this comment; we will add these points to the introduction/discussion sections in order to clarify the possible impact of our work.
>
> (3) By interpolation we mean that the training data is fit exactly, i.e., the training loss is 0.  After training, an overparameterized neural network can interpolate the data, i.e., it  implements a continuous function that perfectly fits the training data. The term interpolation has been used in this way in a number of recent works, e.g. in: https://arxiv.org/abs/1806.05161, https://arxiv.org/abs/1712.06559, https://arxiv.org/abs/1903.08560, https://arxiv.org/abs/1906.11300, https://arxiv.org/abs/1810.07288. We will make sure to clarify this in our paper.
>
> (4) We will change the text to “grayscale”.
>
> (5) You are correct: Since Figure 2b Example 6 is not an attractor, iterating the network on a perturbed version of this image will lead to a different training example.
>
> (6) Thanks for the careful reading of our paper. Figure 2b’s caption is correct as is. Earlier in the paper, we performed one experiment with 10k examples to demonstrate the robustness of this phenomenon.  Since iterating 10k examples in every experiment is too computationally expensive and provides little additional insight, we only performed this large experiment once.
>
> (7) See (3).
>
> (8, 9) Since these are standard practice, we initially felt that it would be sufficient to refer to only one reference (the Deep Learning Book), but agree with the proposed change to add these citations to our paper.
>
> (10) We do not use weight decay in Adam or RMSProp.
>
> (11) See (3).
>
> (12) All of the figures provided are color CIFAR10 images, as we wanted to provide illustrative examples for the figures (these settings only use 10 training examples).  The footnote on page 4 explains our reason for using grayscale images when running experiments on 100 images.
>
> (13) Thank you for pointing this out.
>
> (14) There seemed to be a glitch with the link, but it seems to be ok now.  Please let us know if it is still not working.
>
> Thank you, again, for the careful reading and helpful comments. Please let us know if you have any further questions.

---

> > ### Comment · AnonReviewer1 · 2019-11-13
> > **Follow up**
> >
> > Thank you for all your answers.
> >
> > (2a) I think this conclusion might be broader than what the results actually show. In particular, I am skeptical that this property would hold for non-autoencoders (due to the vastly different amount of mutual information between the inputs and targets), and I am skeptical that this property would hold for amounts of training normally seen in practice. It is interesting I agree that DNNs are such a wide function class that they contain contractive mappings that can be found with AEs in the overparametrized-overtraining regime, but I'd wager this is where this property stops, rather than being a property of all DNNs. Adversarial examples come to mind as evidence of this.
> > As mentioned in the last point of my comments, this paper piqued my curiosity enough that I reimplemented some of the experiments. Getting similar results required amounts of tweaking and waiting that made me skeptical of this contractive property being ubiquitous in DNNs.
> >
> > (2bc) that is indeed intriguing but I am not familiar enough with this literature to judge, and I think in general the ICLR crowd may not be able to appreciate such results.
> >
> > I'll reiterate my sentiment that I think the authors should continue work on this, and that another venue where readers are more likely to be familiar with the biological implications of this phenomenon may appreciate this work more.

---

> > > ### Author Response · Authors · 2019-11-14
> > > **Response to Follow Up**
> > >
> > > (2a) It is not surprising DNNs are expressive enough to learn
> > > contractive maps, since DNN's represent a broad class of functions.
> > >
> > > However, it is surprising that standard gradient descent converges to
> > > such a map (i.e., providing a form of self-regularization).  In
> > > practice, gradient descent may stop short of an actual contractive map,
> > > but this inductive bias is something new and should be of wide interest.
> > >
> > > Moreover, prior works on inductive biases of deep networks often
> > > consider the limit as the training error goes to 0 just as we do: see,
> > > e.g., https://arxiv.org/abs/1705.09280,
> > > https://arxiv.org/pdf/1806.00468.pdf. While this limit may not be
> > > achieved in practical training, it can expose important hidden
> > > properties of these systems.
> > >
> > > (2bc) We respectfully disagree that the ICLR crowd may not be able to
> > > appreciate or would not be interested in these results.  Learning
> > > representations is in the name of the conference, and neuroscience is
> > > specifically mentioned in the call for papers.

---

### Official Review · AnonReviewer3 · 2019-11-03
**Official Blind Review #3**

**Rating:** 3

**Review:**

Summary:
This paper explores the properties of an auto-encoder to behave as an associative memory retrieval mechanism. The authors show really interesting results where they are able to retrieve a small subset of encoded images (mnist) by giving the autoencoder random noise. They also show they can retrieve full videos by giving the autoencoder the output frame from the previous timestep.

The overall problem is a really interesting one which is to try to develop associative memory, retrieval models.

Decision:
Reject

Reasons:
1. Although the work is interesting, the only related work the authors cover is hopfield networks. A cursory search indicates that this has been done before (k. Niki IEEE, Trischler 2016, M.A.Kramer 1992).

Improvement:
1. A more thorough discussion of related work would be helpful.
2. A direct qualitative comparison to related work would also be helpful.

**Experience Assessment:**

I do not know much about this area.

**Review Assessment: Checking Correctness Of Derivations And Theory:**

I did not assess the derivations or theory.

**Review Assessment: Checking Correctness Of Experiments:**

I assessed the sensibility of the experiments.

**Review Assessment: Thoroughness In Paper Reading:**

I read the paper at least twice and used my best judgement in assessing the paper.

---

> ### Author Response · Authors · 2019-11-07
> **Response to Review #3**
>
> We thank the reviewer for the comments and for emphasizing that the problem we consider is interesting.
>
> In writing our paper, we have performed a careful literature review.
>
> The phenomenon that networks trained using standard optimization methods store training examples as attractors or sequences of examples as limit cycles has to the best of our knowledge not been observed before in the literature.
>
> Thank you for the references, but after carefully checking, none of the papers make the observation that training examples become attractors: (1) M.A. Kramer merely introduces and defines autoencoders.  (2) K. Niki is work from 1989 that studies small networks in the binary input setting. (3) Trischler 2016 studies how to train recurrent networks to model dynamical systems.
>
> We hope this addresses your concerns and please let us know if there are any other questions.

---

### Decision · Program_Chairs · 2019-12-19

**Decision:**

Reject

**Comment:**

The paper shows that overparameterized autoencoders can be trained to memorize a small number of training samples, which can be retrieved via fixed point iteration. After rounds of discussion with the authors, the reviewers agree that the idea is interesting and overall quality of writing and experiments is reasonable, but they were skeptical regarding the significance of the finding and impact to the field and thus encourage studying the phenomenon further and resubmitting in a future conference. I thus recommend rejecting this submission for now.